# Exceptional Response to Trastuzumab Deruxtecan (T-DXd) in HER2-Positive Metastatic Endometrial Cancer

**DOI:** 10.3390/curroncol32110596

**Published:** 2025-10-24

**Authors:** Riccardo Vida, Michele Bartoletti, Lucia Lerda, Serena Corsetti, Simona Scalone, Anna Calabrò, Angela Caroli, Monica Rizzetto, Giulia Zapelloni, Elisabetta Caccin, Stefano Fucina, Giorgia Bortolin, Sara Cecco, Paolo Baldo, Sandro Pignata, Daniela Califano, Vincenzo Canzonieri, Antonino Ditto, Fabio Puglisi

**Affiliations:** 1Department of Medicine (DMED), University of Udine, 33100 Udine, Italy; riccardo.vida@cro.it (R.V.); monica.rizzetto@cro.it (M.R.); giulia.zapelloni@cro.it (G.Z.); fabio.puglisi@cro.it (F.P.); 2Unit of Medical Oncology and Cancer Prevention, Department of Medical Oncology, Centro di Riferimento Oncologico di Aviano (CRO), Istituto di Ricovero e Cura a Carattere Scientifico (IRCCS), National Cancer Institute, 33081 Aviano, Italy; serena.corsetti@cro.it (S.C.); sscalone@cro.it (S.S.); 3Pathology Unit, Centro di Riferimento Oncologico di Aviano (CRO), Istituto di Ricovero e Cura a Carattere Scientifico (IRCCS), National Cancer Institute, 33081 Aviano, Italy; lucia.lerda@cro.it (L.L.); vcanzonieri@cro.it (V.C.); 4Nuclear Medicine Unit, Centro di Riferimento Oncologico di Aviano (CRO), Istituto di Ricovero e Cura a Carattere Scientifico (IRCCS), National Cancer Institute, 33081 Aviano, Italy; anna.calabro@cro.it; 5Department of Radiation Oncology, Centro di Riferimento Oncologico di Aviano (CRO), Istituto di Ricovero e Cura a Carattere Scientifico (IRCCS), National Cancer Institute, 33081 Aviano, Italy; angela.caroli@cro.it; 6Clinical Trial Office, Scientific Direction, Centro di Riferimento Oncologico di Aviano (CRO), Istituto di Ricovero e Cura a Carattere Scientifico (IRCCS), National Cancer Institute, 33081 Aviano, Italy; elisabetta.caccin@cro.it; 7Gynecologic Oncology Unit, Centro di Riferimento Oncologico di Aviano (CRO), Istituto di Ricovero e Cura a Carattere Scientifico (IRCCS), National Cancer Institute, 33081 Aviano, Italyantonino.ditto@cro.it (A.D.); 8Pharmacy Unit, Centro di Riferimento Oncologico di Aviano (CRO), Istituto di Ricovero e Cura a Carattere Scientifico (IRCCS), National Cancer Institute, 33081 Aviano, Italy; giorgia.bortolin@cro.it (G.B.); scecco@cro.it (S.C.); pbaldo@cro.it (P.B.); 9Uro-Gynecological Medical Oncology, Istituto Nazionale Tumori, Istituto di Ricovero e Cura a Carattere Scientifico (IRCCS), Fondazione G Pascale, 80131 Naples, Italy; s.pignata@istitutotumori.na.it; 10Microenvironment Molecular Targets Unit, Istituto Nazionale Tumori, Istituto di Ricovero e Cura a Carattere Scientifico (IRCCS), Fondazione G Pascale, 80131 Naples, Italy; d.califano@istitutotumori.na.it; 11Department of Medical, Surgical and Health Sciences, University of Trieste, 34127 Trieste, Italy

**Keywords:** endometrial cancer, targeted therapy

## Abstract

**Simple Summary:**

According to the SEER database, endometrial cancer has shown a steady increase in both incidence and mortality. While patients diagnosed at an early stage generally have a favourable prognosis, outcomes in advanced settings remain poor. Among histological subtypes, serous endometrial carcinoma is associated with particularly poor outcomes. Notably, approximately 35% of serous endometrial cancers exhibit HER2 overexpression, suggesting a potential role for HER2-targeted therapies in this subgroup. However, HER2 status is not routinely assessed in clinical practice, as it is not included in the standard diagnostic workflow recommended by the TCGA/ProMisE classification. This case report is of particular interest as it describes a rare clinical scenario involving coexistence of dMMR, serous histology, and HER2 overexpression, which is generally uncommon in the same tumour. For this small but clinically relevant subset of patients, optimal treatment sequencing and prioritization remain challenging, highlighting the need for further investigation into tailored therapeutic strategies as well as comprehension of resistance mechanisms.

**Abstract:**

**Objectives:** Endometrial cancer is the most common gynaecologic malignancy, and its mortality rate is rising. Advanced or recurrent disease remains challenging because historically there have been limited therapeutic options. We aim to describe a complete and durable response to the HER2-directed antibody–drug conjugate trastuzumab deruxtecan (T-DXd) in a heavily pretreated patient with HER2-positive, mismatch-repair-deficient metastatic serous endometrial cancer. **Methods**: A 72-year-old woman underwent hysterectomy, bilateral salpingo-oophorectomy, and staging procedures for FIGO stage IIIA, high-grade serous papillary endometrial carcinoma. Tumour profiling revealed dMMR, a p53 abnormal pattern, and HER2 overexpression (IHC 3+). She received carboplatin/paclitaxel plus avelumab, followed by pegylated liposomal doxorubicin and weekly paclitaxel. After progression on paclitaxel, off-label T-DXd was initiated. Molecular data (FoundationOne CDx) were collected, along with and serial imaging and CA125 assessments. **Results**: The patient developed cough after two cycles of T-DXd; interstitial lung disease was excluded, and treatment resumed with steroid cover. By December 2024, PET/CT demonstrated complete metabolic response, with resolution of vaginal-vault and para-aortic lesions and normalisation of CA125. Real-world progression-free survival exceeded eight months, with ongoing symptom improvement. Treatment was generally well tolerated; the principal adverse event was grade 3 neutropenia requiring dose reduction. No cardiotoxicity or interstitial lung disease occurred. **Conclusions**: This case illustrates that T-DXd can induce deep and durable remission in HER2-positive, dMMR metastatic serous endometrial cancer after multiple lines of therapy. It adds real-world evidence supporting further investigation of HER2-directed antibody–drug conjugates in gynaecologic malignancies, and underscores the need for confirmatory trials and refined biomarker-driven patient selection.

## 1. Introduction

While early-stage endometrial cancer generally has a favourable prognosis, with 5-year overall survival (OS) exceeding 80%, patients with recurrent or metastatic disease face poor outcomes, with median OS around 30 months [1]. Recent advancements, such as immunotherapy alone or in combination with chemotherapy or antiangiogenics, have improved in patients with deficient mismatch repair/microsatellite instability-high (dMMR/MSI-h) tumours. However, a substantial clinical need remains, particularly in post-immunotherapy settings and among patients with proficient mismatch repair (pMMR) tumours, who represent approximately 70% of cases and derive limited benefit from immunotherapy [2]. Consequently, novel targeted therapies, including antibody–drug conjugates (ADCs), are under investigation, with promising results in early-phase trials.

## 2. Case Presentation

A 72-year-old woman presented with vaginal bleeding. Hysteroscopy with biopsy in September 2018 confirmed high-grade serous papillary endometrial adenocarcinoma. A staging CT scan of the thorax and abdomen showed no metastases.

She underwent hysterectomy with bilateral salpingo-oophorectomy, pelvic lymph node dissection, and infracolic omentectomy. Pathology revealed synchronous left ovarian involvement, with a definitive stage of pT3aN0, FIGO IIIA according to the 2009 classification. A molecular report returned a dMMR profile with a concurrent p53 abnormal immunohistochemical pattern and an overexpression of HER2 defined as 3+ by IHC according to gastric criteria. The following patient’s disease course with treatments is illustrated in Figure 1.

Postoperatively, a PET/CT identified a solid mass in the right iliac fossa. Considering the aggressive histology and the disease stage, and given her good performance status despite cardiovascular and metabolic comorbidities, she was enrolled in the MITO-END3 trial and allocated to the experimental arm with carboplatin/paclitaxel plus avelumab [3,4]. As part of the clinical trial, a FoundationONE CDx test was performed on tissue biopsy: molecular analyses confirmed the MSI-high status and a high tumour mutational burden (13 Muts/Mb); PD-L1 and PD-L2 genes were also found to be amplified. Among single genes analysed, mutation of TP53 (with a VAF of 63.9%) was detected, in line with the aggressive serous histology; however, no PIK3CA, AKT, or PTEN alterations were identified. No specific druggable mutations were reported.

Due to recurrent neutropenia, the doses of carboplatin and paclitaxel were reduced to 50% and 75%, respectively. After the conclusion of the induction chemo-immunotherapy phase, radiological disease progression was documented on CT scan due to the appearance of a new pulmonary nodule. Therefore, the patient’s treatment within the trial protocol was discontinued, and she was proposed a second-line chemotherapy regimen with pegylated liposomal doxorubicin which was administered from 17 July 2019 to 30 April 2020. The response to this therapy was good. After 10 infusions, treatment was withheld due to the partial response being achieved and because the maximum cumulative dose of anthracycline had been delivered. The patient was subsequently followed with CT scans and clinical evaluations, achieving a duration of response of nearly three years, with stability of the multiple lymphadenopathies along the right iliac vessels.

However, a CT scan on 10 February 2023 described disease progression on the right side of the vaginal vault; this was also confirmed by PET/CT. Given these findings, a third-line treatment with weekly paclitaxel on days 1, 8, and 15 every 28 days was proposed.

Upon confirmation of disease progression from the CT scan of 21 February 2024, and given the limited therapeutic options, the prior treatment with the anti-PD-L1 avelumab, and the lack of benefit from IO-based therapy, a subsequent off-label treatment with T-DXd was proposed, based on the results from the endometrial cohort in the DESTINY-PanTumor-02 [5].

Treatment with the ADC was started. After the first two cycles of T-DXd, a HRCT scan was required due to the onset of cough, and steroid treatment with prednisone 50 mg daily was prescribed. Following an internal radiological review of the CT scan images, interstitial lung disease was excluded, and treatment therefore resumed.

The subsequent cycles were well tolerated, but due to recurrent grade 3 neutropenia events, a dose reduction was implemented. In December 2024, the first radiological evaluation via PET/CT showed a complete response, with disappearance of the previously described nodule at the vaginal vault, lesions located cranially to the previous lesion in the left paramedian region, and lymph node findings along the para-aortic chains (Figure 2). Concomitantly, a rapid normalization of CA125 levels was also documented.

In addition to clinical evaluation, ECG and echocardiography were performed at baseline and every 3–4 months subsequently. Interstitial lung disease was monitored using high-resolution computed tomography (HRCT) at baseline, after 14 weeks, and then every 5–6 months, in accordance with clinical practice. No signs of cardiac or pulmonary toxicity were observed.

## 3. Discussion

This case illustrates an exceptional response to T-DXd in a patient who had received three prior lines of treatment including anti-PD-L1 inhibitor. A real-world progression-free survival of about eight months was achieved, along with improvement in symptom control and an easily manageable toxicity profile. These results are remarkable, given the lack of treatment options for patients with recurrent serous endometrial cancer after failure of platinum salts, immune-check point inhibitors, anthracyclines, and taxanes.

After revolutionizing the treatment algorithm for HER2-positive advanced breast and gastric cancers, T-DXd has shown impressive activity in solid tumours overexpressing HER2. In the case of DESTINY-PanTumour-02, an impressive overall response rate (ORR) of 57.5% was observed in a cohort of HER2-expressing endometrial cancers (40 patients), with ORR reaching 84.6% in IHC 3+ tumours [5]. This led to approval from the FDA but not the EMA (which cited the lack of a confirmatory phase III trial), therefore limiting prescription of T-DXd in Europe for these patients.

In our case, HER2 3+ expression was identified in approximately 20% of tumour cells. Despite this heterogeneous pattern, a complete response was achieved, with complete disappearance of target lesion on PET/CT. This observation further supports the established concept of the bystander killing effect of T-DXd, whereby therapeutic activity is observed even in tumours with limited HER2 overexpression (above the 10% cut-off) [6,7]. These findings raise questions regarding the adequacy of the current HER2 scoring system—originally developed to predict response to trastuzumab—for guiding the use of HER2-targeted ADCs [8,9]. Although this scoring system can effectively predict therapeutic response (as seen in our case) alternative approaches, such as the H-score system employed in exploratory analyses of other ADCs, may provide improved predictive value [10,11]. Moreover, integrating digital pathology and artificial intelligence could, in the future, yield more precise and individualized criteria for optimizing patient selection for HER2-directed therapies.

Initial therapy selection in this case was guided by dMMR status, consistent with TCGA-driven molecular classification, which prioritizes immune checkpoint inhibitors in dMMR tumours. Therefore, the patient was enrolled in the trial, providing her with access to the potentially most effective treatment based on the disease characteristics. Though IO-based therapy was unavailable at the time, it did have a strong biological rationale which was later confirmed by registrational trials. As a result, trastuzumab combined with platinum-based chemotherapy was not administered in the first-line setting.

Data from a relatively small phase II trial have demonstrated the potential benefit of such an approach [12], and a confirmatory phase III study, NRG-GY026, is currently ongoing [13]. However, the decision to start with immunotherapy is now supported by several phase III trials which have shown significant benefits from such a strategy in patients with dMMR endometrial cancer, with improvements observed in both progression-free survival (PFS; HR 0.34, 95% CI 0.27–0.44) [2] and overall survival. These findings led to the establishment of this approach—experimental at the time of the case described—as the current standard of care.

Contrary to expectations, however, avelumab demonstrated limited efficacy in this case, with a PFS of only 6 months and disease progression (PD) occurring even before initiation of maintenance therapy with the immune checkpoint inhibitor. When evaluating the clinical trials that supported approval of first-line IO-based strategies in endometrial cancer, it is revealed that combination of IO and chemotherapy showed no significant difference in efficacy for serous histology compared to the intent-to-treat (ITT) population. A less consistent benefit for this histological subtype was reported only in the NRG-GY018 trial. Notably, serous histology was predominantly observed in the pMMR subgroup, while it was rare among patients with dMMR (5, 2, and 0 patients in the RUBY, NRG-GY018, and ATTEND trials, respectively) [14,15,16].

Moreover, although dMMR is considered a more reliable and robust biomarker than PD-L1, it is not without limitations, and it must be acknowledged that approximately 20% of patients with dMMR tumours exhibit primary resistance to immunotherapy. Similarly, in colorectal cancer, where dMMR status also guides IO treatment, not all patients derive clinical benefit from immunotherapy [17,18]. This highlights the need for further refinement of predictive biomarkers and a deeper understanding of resistance mechanisms.

Although uncommon and, consequently, underrepresented in the literature, the coexistence of serous histology (with TP53 alteration and/or HER2 overexpression) and dMMR profile can occur. Whether treatment should be prioritized based on MMR status in this subgroup remains unclear, as most data on dual-classifier tumours pertain to endometrioid histology. According to the TCGA classification of endometrial cancer, a hierarchical testing approach is required [19], with treatment prioritized based on molecular profiling [20,21]. In our case, the tumour exhibited both a dMMR and a p53 abnormal profile (the latter being frequently associated with HER2-positive status in serous histology) thus supporting the decision to initiate an immunotherapy-based regimen.

Drawing comparisons with other HER2-driven malignancies, opposite results have been reported: in gastric cancer—despite relying on PD-L1 as a biomarker rather than MMR status—significant results have been reported in first-line treatment, with pembrolizumab and trastuzumab becoming the standard of care following the results of the KEYNOTE-811 trial [22]; conversely, in breast cancer, the addition of immunotherapy to HER2-directed treatment has not demonstrated clinical benefit in either early-stage disease [23]. As for endometrial cancer, HER2 status is not currently integrated into the TCGA molecular classification. Given the availability of effective targeted therapies, the aggressive nature of serous endometrial carcinoma, and the strong predictive value of HER2 as a biomarker, it is essential to define its role within the current diagnostic and treatment framework to further refine therapeutic strategies for this challenging subset of patients.

## 4. Conclusions

This case documents a complete and durable response to trastuzumab deruxtecan, in a patient with HER2-positive, dMMR, metastatic endometrial cancer, after multiple prior therapies. It supports further investigation of HER2-directed ADC in gynaecologic malignancies and highlights the urgent need for confirmatory phase III trials and broader access to these agents.

## Figures and Tables

**Figure 1 curroncol-32-00596-f001:**
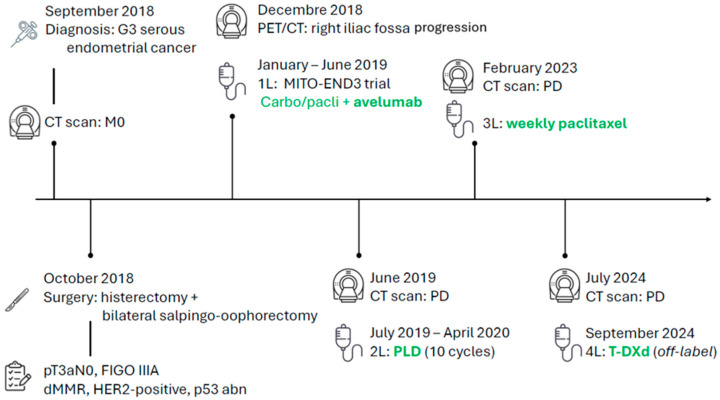
Patient journey timeline.

**Figure 2 curroncol-32-00596-f002:**
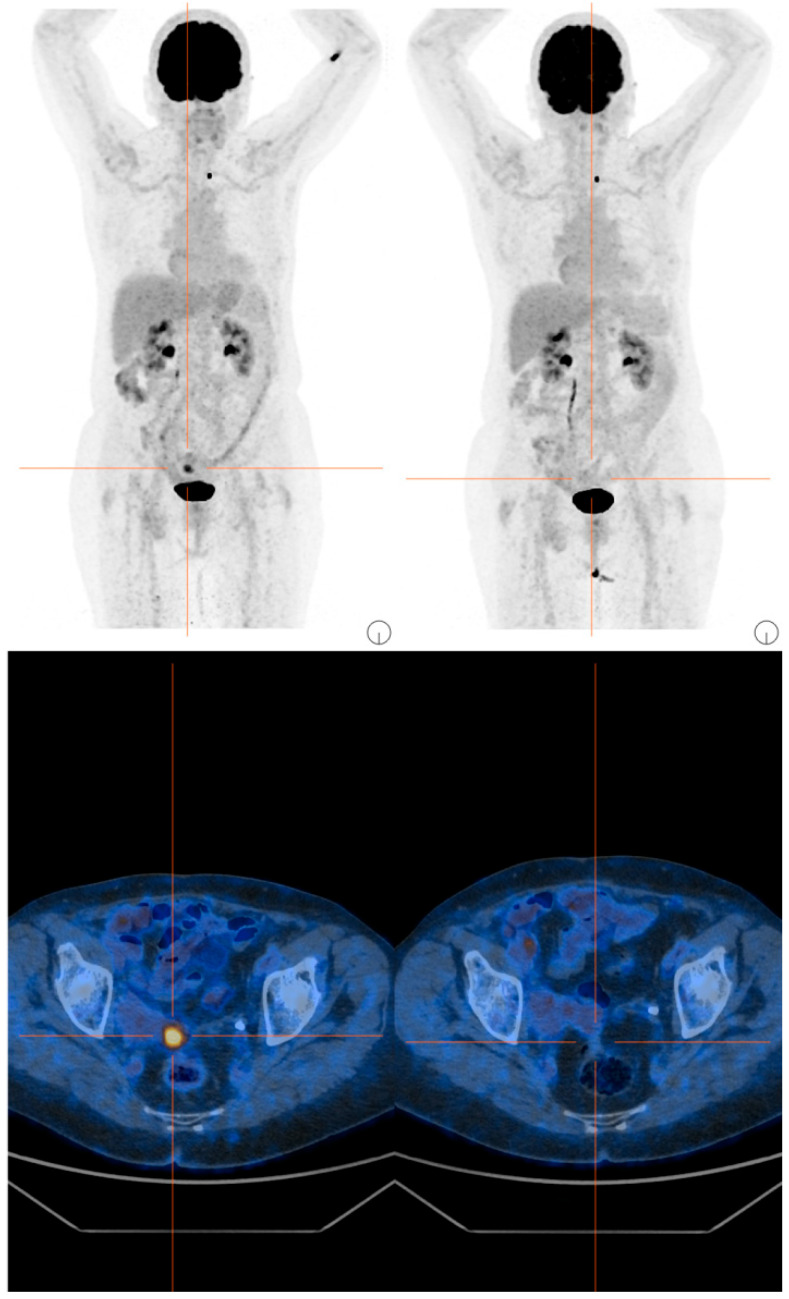
PET/CT scans before and after 4 cycles of T-DXd.

## Data Availability

The dataset supporting the conclusions of this article is included in this article.

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
