# Peer review of "Exceptional Response to Trastuzumab Deruxtecan (T-DXd) in HER2-Positive Metastatic Endometrial Cancer"

_curroncol, 2025, doi:10.3390/curroncol32110596_

Round 1

Reviewer 1 Report

Comments and Suggestions for Authors

The manuscript presents a case report describing an exceptional and durable response to trastuzumab deruxtecan (T-DXd) in a heavily pretreated patient with HER2-positive, dMMR serous endometrial carcinoma. The case is clinically relevant, timely, and adds to the growing evidence supporting HER2-targeted antibody–drug conjugates in gynecologic malignancies. Presentation of the patient and related discussion are quite appropriate for the scope of the manuscript. I have some suggestions that I feel could improve the quality of the manuscript.

1- A short literature summary (prior case reports, sub-analyses from DESTINY-PanTumor02, and real-world experiences) can be tabulated.

2- Please expand on the discussion of how this case contributes to the ongoing debate about HER2 testing and scoring in gynecologic malignancies, particularly the importance of heterogeneous HER2 expression.

3- Was the gastric cancer HER2 scoring used throughout, or were gynecology-specific adjustments applied?

4- The patient progressed rapidly after ICI-based therapy. Please expand on the discussion of resistance mechanisms in dMMR tumors, particularly in the context of serous histology.

5- The article states that interstitial lung disease (ILD) was excluded, but the exclusion methodology (radiology, pulmonology evaluation, HRCT timing) should be specified.

6- Minor typographical issues (e.g., “guided by was guided by dMMR status” should be corrected).

Comments on the Quality of English Language

The English could be improved to more clearly express the research.

Author Response

1- A short literature summary (prior case reports, sub-analyses from DESTINY-PanTumor02, and real-world experiences) can be tabulated.

Answer

With regard to this issue, the data of ORR from the DESTINY-PanTumor02 trials were discussed at lines 152-154 (new version uploaded)

2- Please expand on the discussion of how this case contributes to the ongoing debate about HER2 testing and scoring in gynaecologic malignancies, particularly the importance of heterogeneous HER2 expression.

3- Was the gastric cancer HER2 scoring used throughout, or were gynaecology-specific adjustments applied?

Answer

About points 2 and 3, HER2 expression was evacuate according to gastric criteria, as requested in the DESTINTY-PanTumor02. This information was already reported in lines 72-74

Regarding the optimal scoring system, we are aware about the current debate, and we have emphasised this in lines 157-169 (new version uploaded)

4- The patient progressed rapidly after ICI-based therapy. Please expand on the discussion of resistance mechanisms in dMMR tumours, particularly in the context of serous histology.

Answer

The rapid progression observed at the first radiological evaluation has already been addressed in lines 165–174, with specific reference to endometrial cancer with serous histology. Defining resistance mechanisms to immunotherapy in this context remains challenging, as the co-occurrence of a dMMR profile and serous histology is rare, as discussed in the case report.

The KEYNOTE-775 trial included both dMMR and pMMR patients; however, subgroup data for serous histology are only reported within the overall population, without specific analysis for the dMMR or pMMR subgroups.

Regarding first-line trials, the small number of enrolled patients with serous histology is acknowledged in the manuscript and does not allow for definitive conclusions.

IO monotherapy in the dMMR population in the second-line setting has been investigated, among others, in three main trials:

  1. KEYNOTE-158: Although 90 patients were enrolled, no specific data were provided on histological subtypes or on potential mechanisms of resistance in patients who experienced progressive disease (PD) as their best response.
  2. GARNET: Response was analysed also in relation to tumour mutational burden (TMB) and PD-L1 expression; however, no details were given on mechanisms of resistance, despite 36.2% of patients showing PD as best response.
  3. PHAEDRA: No patients with serous histology were included in this trial.

5- The article states that interstitial lung disease (ILD) was excluded, but the exclusion methodology (radiology, pulmonology evaluation, HRCT timing) should be specified.

Answer

Regarding point 5, outside a clinical trial, monitoring was performed in line with current clinical practice and in order not to impact too much on QoL of our patient.

Additional information has been added in lines 136-140 (new version uploaded)

6- Minor typographical issues (e.g., “guided by was guided by dMMR status” should be corrected)

Answer

We have corrected the mistake at line 170

Reviewer 2 Report

Comments and Suggestions for Authors

Dear Authors thank for your effort.

According to your timeline; Pet-CT at December 2018 showed tumor in right iliac fossa, So, It is progressive disease not recurrence.  Also the patients had been treated Carbo/Pacli+ avelumab at between January-June 2019 and CT scan showed PD at June 2019. In discussion section, you have descripted 6 months PFS following this combination. Will you explain these two problems.  

Author Response

According to your timeline, Pet-CT at December 2018 showed tumor in right iliac fossa, So, It is progressive disease not recurrence.  Also, the patients had been treated Carbo/Pacli+ avelumab at between January-June 2019 and CT scan showed PD at June 2019. In discussion section, you have descripted 6 months PFS following this combination. Will you explain these two problems.

Answer

Regarding the first issue, the image has been corrected.

Regarding the second point, from January to June elapse 6 months, so was defined PFS. According to the trial design of the MITO-END3 trial, PFS was calculated from start of chemotherapy and not from maintenance initiation.

Reviewer 3 Report

Comments and Suggestions for Authors

Summary

This presented study reported a patient with stage IIIA high-grade serous papillary endometrial cancer characterized by HER2-positive, dMMR, and abnormal p53 expression. The patient underwent hysterectomy and bilateral salpingo-oophorectomy, followed by multiple lines of systemic therapy. After progression on paclitaxel, off-label treatment with T-DXd was initiated and the patient showed remarkable response, supporting further investigation of HER2-directed antibody–drug conjugates in gynecologic malignancies. Overall, the authors described very detailed chronology. However, considering that prior reports (PMID: 37595181, 37325293) and an ongoing clinical trial (NCT06989112) have already described the potential application of T-DXd in HER2-positive endometrial cancer, the authors should further clarify the specific novelty and distinguishing features of this case

Below are my suggestions for improvement-

  1. The current introduction reads more like a general background. The authors should highlight the novelty and clearly explain why this particular case is worth reporting. I recommend that the authors expand the Introduction to include: 1) a brief introduction of HER2 overexpression in endometrial carcinoma, especially in the serous subtype, since HER2 testing is not standard in endometrial cancer; 2) a rationale for reporting this case; and 3) a statement describing what distinguishes this case from previous reports.
  2. The Result section included two figures, but they are not referenced in the main text. To improve readability and ensure that the figures support the case narrative, I recommend integrating in-text citations (ex. “Figure 1”, “Figure 2”) at appropriate points in the Results section when the corresponding data.
  3. Precision terminology: The Result section used “deep partial response” and “outstanding response/disappearance”, which are not standard clinical terms and may be ambiguous. Please consider revising them for better accuracy, preferably RECIST/PERCIST terminology.

Author Response

This presented study reported a patient with stage IIIA high-grade serous papillary endometrial cancer characterized by HER2-positive, dMMR, and abnormal p53 expression. The patient underwent hysterectomy and bilateral salpingo-oophorectomy, followed by multiple lines of systemic therapy. After progression on paclitaxel, off-label treatment with T-DXd was initiated and the patient showed remarkable response, supporting further investigation of HER2-directed antibody–drug conjugates in gynaecologic malignancies. Overall, the authors described very detailed chronology. However, considering that prior reports (PMID: 37595181, 37325293) and an ongoing clinical trial (NCT06989112) have already described the potential application of T-DXd in HER2-positive endometrial cancer, the authors should further clarify the specific novelty and distinguishing features of this case

Answer

We are aware of the ongoing clinical trial evaluating first-line treatment with T-DXd in patients with HER2-positive endometrial cancer. However, both data availability and regulatory approval are not expected in the immediate future. In this context, we believe it is important to highlight and reinforce the efficacy data already emerging from real-world settings.

Regarding PMID: 37595181, while three cases were reported, they were only briefly described from a molecular perspective. In contrast, our report provides additional information derived from next-generation sequencing (NGS) analysis. Moreover, in all our cases, patients presented with mismatch repair proficient (pMMR) tumors.

As for PMID: 37325293, although it represents an interesting case report, the treatment approach differs significantly from standard clinical practice in Europe. In our case report, the patient received immunotherapy as first-line treatment within a clinical trial setting, which aligns with the current standard of care for patients with advanced endometrial cancer. Furthermore, while the patient in PMID: 37325293 had a pMMR/MSS tumor, a distinctive feature of our case is the co-occurrence of pMMR status, HER2 overexpression, and serous histology.

Below are my suggestions for improvement-

The current introduction reads more like a general background. The authors should highlight the novelty and clearly explain why this particular case is worth reporting. I recommend that the authors expand the Introduction to include: 1) a brief introduction of HER2 overexpression in endometrial carcinoma, especially in the serous subtype, since HER2 testing is not standard in endometrial cancer; 2) a rationale for reporting this case; and 3) a statement describing what distinguishes this case from previous reports.

Answer

We thank you for this suggestion. In accordance with the publisher's request, we have included a Simple Summary in which we incorporated this information.

The Result section included two figures, but they are not referenced in the main text. To improve readability and ensure that the figures support the case narrative, I recommend integrating in-text citations (ex. “Figure 1”, “Figure 2”) at appropriate points in the Results section when the corresponding data.

Answer

We thank the reviewer for the valuable suggestion. We have revised and improved the manuscript accordingly.

Precision terminology: The Result section used “deep partial response” and “outstanding response/disappearance”, which are not standard clinical terms and may be ambiguous. Please consider revising them for better accuracy, preferably RECIST/PERCIST terminology.

Answer

We thank the reviewer for the suggestion. We have revised and improved the manuscript accordingly.

Round 2

Reviewer 1 Report

Comments and Suggestions for Authors

I am satisfied that the authors have addressed all of my previous concerns about the article. It is now much improved and I feel that it is now suitable for publication.

Comments on the Quality of English Language

The English could be improved to more clearly express the research.